# Ubiquinol Ameliorates Endothelial Dysfunction in Subjects with Mild-to-Moderate Dyslipidemia: A Randomized Clinical Trial

**DOI:** 10.3390/nu12041098

**Published:** 2020-04-15

**Authors:** Jacopo Sabbatinelli, Patrick Orlando, Roberta Galeazzi, Sonia Silvestri, Ilenia Cirilli, Fabio Marcheggiani, Phiwayinkosi V. Dludla, Angelica Giuliani, Anna Rita Bonfigli, Laura Mazzanti, Fabiola Olivieri, Roberto Antonicelli, Luca Tiano

**Affiliations:** 1Department of Clinical and Molecular Sciences, Università Politecnica delle Marche, Via Tronto 10/A, 60126 Ancona, Italy; 2Department of Life and Environmental Sciences, Università Politecnica delle Marche, Via Ranieri 65, 60128 Ancona, Italy; 3Clinical Laboratory and Molecular Diagnostics, IRCCS INRCA, Via della Montagnola 81, 60127 Ancona, Italy; 4Biomedical Research and Innovation Platform, South African Medical Research Council, P.O. Box 19070, Tygerberg, South Africa; 5Scientific Direction, IRCCS INRCA, Via della Montagnola 81, 60127 Ancona, Italy; 6Department of Clinical Sciences, Biology and Biochemistry Section, Università Politecnica delle Marche, Via Ranieri 65, 60128 Ancona, Italy; 7Center of Clinical Pathology and Innovative Therapy, IRCCS INRCA, 60121 Ancona, Italy; 8Cardiology Unit, IRCCS INRCA, Via della Montagnola 81, 60127 Ancona, Italy

**Keywords:** coenzyme Q10, endothelial dysfunction, flow-mediated dilation, oxidized LDL, dyslipidemia

## Abstract

In this randomized, double-blind, single-center trial (ANZCTR number ACTRN12619000436178) we aimed to investigate changes in endothelium-dependent vasodilation induced by ubiquinol, the reduced form of coenzyme Q10 (CoQ10), in healthy subjects with moderate dyslipidemia. Fifty-one subjects with low-density lipoprotein (LDL) cholesterol levels of 130–200 mg/dL, not taking statins or other lipid lowering treatments, moderate (2.5%–6.0%) endothelial dysfunction as measured by flow-mediated dilation (FMD) of the brachial artery, and no clinical signs of cardiovascular disease were randomized to receive either ubiquinol (200 or 100 mg/day) or placebo for 8 weeks. The primary outcome measure was the effect of ubiquinol supplementation on FMD at the end of the study. Secondary outcomes included changes in FMD on week 4, changes in total and oxidized plasma CoQ10 on week 4 and week 8, and changes in serum nitrate and nitrite levels (NOx), and plasma LDL susceptibility to oxidation in vitro on week 8. Analysis of the data of the 48 participants who completed the study demonstrated a significantly increased FMD in both treated groups compared with the placebo group (200 mg/day, +1.28% ± 0.90%; 100 mg/day, +1.34% ± 1.44%; *p* < 0.001) and a marked increase in plasma CoQ10, either total (*p* < 0.001) and reduced (*p* < 0.001). Serum NOx increased significantly and dose-dependently in all treated subjects (*p* = 0.016), while LDL oxidation lag time improved significantly in those receiving 200 mg/day (*p* = 0.017). Ubiquinol significantly ameliorated dyslipidemia-related endothelial dysfunction. This effect was strongly related to increased nitric oxide bioavailability and was partly mediated by enhanced LDL antioxidant protection.

## 1. Introduction

Dyslipidemia, i.e., the alteration of one or more blood lipid fractions, including total cholesterol (TC), low-density lipoprotein (LDL)-cholesterol (LDL-C), triglycerides, and high-density lipoprotein (HDL)-cholesterol (HDL-C), is a major risk factor for cardiovascular disease (CVD) [1]. Indeed, a strong and graded association exists between the levels of TC and LDL-C and CVD mortality at all blood pressure levels [2]. It is estimated that dyslipidemia has a prevalence of 38.6% among individuals aged 40 years and above [3], and the alteration of TC, HDL-C, or non-HDL-C has been reported in one out of five US children and adolescents aged 8–17 years [4].

A well-established connection exists between cardiovascular risk factors, including dyslipidemia, endothelial dysfunction (ED), and atherosclerosis in the onset of cardiovascular disease [5,6]. The vascular endothelium plays a pivotal role in vascular homeostasis, sensing circulating signals that are capable of inducing phenotypic alterations of vessel walls and releasing a variety of autocrine and paracrine substances [7,8]. The endothelium modulates the vascular tone mainly through nitric oxide (NO) synthesis by the endothelial isoform of NO synthase (eNOS) [9,10]. The shear stress exerted on the endothelium by flowing blood induces NO synthesis and release into the tunica media [11].

The impairment of endothelial function (EF) predates the morphological changes of atherosclerosis and can mechanistically contribute to atherosclerosis-related diseases [12,13,14]. Current approaches to measure endothelial vasomotor function involve transient physiological or pharmacological endothelium stimulation and recording of responses [15]. Flow-mediated dilation (FMD) assumes that ED is a systemic process involving the coronary arteries as well as the peripheral circulation. It estimates ultrasonographically the dilation of a large peripheral conduit artery, typically the brachial artery, in response to the increased blood flow resulting from the removal of a transient ischemic stimulus. FMD has proved effective in evaluating the impact of several interventions on ED [16,17,18]. However, poor reproducibility related to operator experience, subject preparation, environmental variables, equipment and image acquisition and analysis [19] prevents its broader clinical use.

Coenzyme Q10 (CoQ10) is a lipophilic endogenous quinone playing a pivotal role in mitochondrial bioenergetics [20]. In its reduced form, i.e., ubiquinol, it is also endowed with antioxidant activity [21]. In particular, ubiquinol exerts a major antioxidant activity in the vascular compartment by preventing LDL oxidation [22]. More recently, it has been shown that beyond these mechanisms, ubiquinol is able to affect cellular biochemistry also by modulating gene expression and by promoting anti-inflammatory effects [20]. These functions provide a strong rationale for its clinical use to treat cardiovascular disease (CVD). CoQ10 has improved endothelium-dependent vasodilation, as measured by FMD, in patients with type 2 diabetes [23,24], or coronary artery disease (CAD) [25]. However, the evidence of its effect on ED in subjects without clinical manifestations of atherosclerosis-related disease is limited. It has been reported that ubiquinone, the oxidized form of CoQ10, was not able to improve EF in healthy subjects with cardiovascular risk factors, in contrast to what has been observed in the secondary prevention setting [25,26]. Given the significantly higher bioavailability of orally administered ubiquinol compared with ubiquinone [27] and its direct antioxidant activity, this study assesses whether 8-week ubiquinol supplementation enhances endothelium-dependent vasodilation in adults with moderate untreated dyslipidemia and without evidence of CVD.

## 2. Materials and Methods 

### 2.1. Participants

The study included men and post-menopausal women aged 35–65 years. Menopause was defined as >12 months without menses. Premenopausal women were excluded due to their generally lower cardiovascular risk according to conventional algorithms [28], and to avoid any source of bias related to the high variability of FMD during the different phases of the menstrual cycle [29]. The inclusion criteria were a body mass index (BMI) between 18.5 and 29.9 kg/m^2^; plasma LDL-C between 130 and 200 mg/dL measured within 1 month prior to the screening visit; and an FMD between 2.5% and 6% at the screening visit. The LDL-C inclusion criteria was defined according to the current European Society of Cardiology (ESC) guidelines to include subjects with established dyslipidemia that could benefit from lifestyle interventions as an alternative to lipid-lowering drugs [28]. Subjects with type 1 or type 2 diabetes, liver disease (serum transaminases or total bilirubin exceeding the upper limit of the laboratory reference range), renal failure (estimated glomerular filtration rate, <60 mL/min), thyroid disorders, CAD, severe malabsorption or inflammatory bowel disease (e.g., Crohn’s disease, ulcerative colitis), severe psychiatric disease, and current smokers were excluded. Those taking lipid-lowering drugs or supplements, including niacin, omega-3 fatty acids, and red yeast rice extract, phosphodiesterase inhibitors, and NO donors were also excluded. The complete list of inclusion and exclusion criteria is provided in Appendix A. The Consolidated Standards of Reporting Trials (CONSORT) flow chart is provided in Figure 1.

### 2.2. Study Design

This was a double-blind, randomized, placebo-controlled, parallel group trial conducted at the hospital facilities of the Italian National Research Center on Aging (INRCA) IRCCS in Ancona, Italy, from December 2016 to June 2017. Based on the pharmacokinetics of ubiquinol [30], a study duration of 8 weeks was chosen to observe a tangible biological effect. Participants provided their written informed consent before enrollment. The study was approved by the Regional Institutional Review Board (Comitato Etico Regione Marche, CERM; approval number 2016-0614 IN) and was conducted in accordance with the ethical principles of the Declaration of Helsinki. The study is registered with the Australia and New Zealand Clinical Trials Registry (ANZCTR) number: ACTRN12619000436178. The study included a screening visit and then evaluations at 4 weeks (T1) and 8 weeks (T2). The full trial protocol is accessible through the ANZCTR website.

Participants were recruited from subjects referred as outpatients to the Clinical Laboratory of INRCA IRCCS and selected according to the above-listed inclusion and exclusion criteria. Those meeting the FMD criteria were enrolled and randomized according to a computer-generated randomized code into three groups who received: (i) ubiquinol 200 mg/day; (ii) ubiquinol 100 mg/day; and (iii) placebo. Dosages were chosen to reflect either the highest amount allowed in Italy and in several EU countries as a nutritional supplement (200 mg), and the amount of ubiquinol contained in many pharmacological formulations (100 mg). The three groups were matched for age, gender, LDL-C, and baseline FMD.

Both the investigators and the patients were blind to the study aims and procedures. A pharmacy designate independent of the study conducted the product dispensing procedure. Each participant received two bottles containing either 100 or 200 mg ubiquinol (Kaneka QH™, Kaneka, Osaka, Japan), the reduced form of CoQ10, which was kindly supplied by Kaneka Corporation (Osaka, Japan); the placebo was in the form of soft gel capsules. The products were self-administered at home, one capsule in the morning and one in the evening for 8 weeks, with food.

At T1 and T2, participants were requested to report any adverse event and underwent a short physical examination and collection of a venous blood sample (approximately 10 mL) to determine plasma CoQ10 and the secondary endpoints. Blood pressure (triplicate, first discarded) and FMD were measured after acclimatization and rest for at least 30 min. 

The intention-to-treat (ITT) population included all randomized subjects, whereas the per-protocol population (PP) included all randomized subjects who completed the 8-week study with only minor protocol violations or deviations.

### 2.3. Primary and Secondary Endpoints

The primary outcome of the study was change in endothelium-dependent vasodilation on week 8 as assessed through non-invasive ultrasound measurement of the FMD at the brachial artery.

Secondary outcomes were: (i) change in FMD at T1 (week 4) and (ii) change in the ratio of oxidized to total plasma CoQ10 at T1 and T2 (week 8); (iii) change in plasma NO, quantified in terms of its stable reaction products (nitrites and nitrates); and (iv) changes in plasma lipoprotein susceptibility to oxidation at T2.

### 2.4. Laboratory Measurements

Serum and plasma samples were stored at −80 °C until analysis. Total white blood cell (WBC), monocyte, and platelet counts were obtained by standard automated procedures. The erythrocyte sedimentation rate (ESR) was measured by the modified Westergren method. Plasma concentrations of total, LDL-C, and HDL-C, triglycerides, creatinine, fasting glucose and insulin, creatine kinase, alanine aminotransferase (ALT), and highly sensitive C-reactive protein (hs-CRP) were measured at both time points using standard procedures. Homeostasis model assessment (HOMA) index was calculated by the formula fasting insulin (μU/mL) × fasting glucose (mg/dl) / 405 [31]. Plasma oxLDL was evaluated through a standard commercial enzyme-linked immunoassay (Mercodia, Uppsala, Sweden). The assay uses the 4E6 mouse monoclonal antibody, detecting an oxLDL particle when its ApoB-100 protein has at least 60 lysine residues substituted with aldehydes due to oxidative modifications. At this level of oxidation, ApoB-100 protein undergoes a conformational change to which the antibody has specificity [32].

### 2.5. Assessment of Flow-Mediated Dilation

FMD was measured as the percent increase of the diameter of the brachial artery in response to increased blood flow due to removal of an ischemic stimulus, i.e., inflation of the sphygmomanometer cuff to 300 mmHg for 5 min. Measurements were taken in fasting subjects who had been resting in supine position for at least 30 min in a quiet room at 20–22 °C using a GE Vivid 7 (GE Healthcare, Holten, Norway) ultrasound scanner. The brachial artery was scanned longitudinally between 5 and 10 cm above the elbow with a 7.5 MHz linear probe held in place by a stereotactic probe holder. The sphygmomanometer cuff was wrapped around the subject’s forearm just below the elbow. Brachial artery diameter was measured on a real-time basis by an automatic edge detection system (FMD studio, Quipu, Pisa, Italy) acquiring simultaneous live duplex ultrasound images. After continuous recording of the baseline brachial artery diameter for 1 min, the cuff was inflated. Post-deflation diameter was monitored continuously for 5 min. FMD was expressed as a percent ratio of the maximum increment of the brachial artery diameter in the post-deflation phase to the baseline diameter. All measurements were taken by the same trained sonographer and assessed by a blinded observer at the end of the study in accordance with current guidelines [33]. Both the sonographer and the observer were blinded to treatment allocation.

### 2.6. Determination of Plasma Coenzyme Q10 Levels

The plasma levels of the oxidized (ubiquinone) and reduced (ubiquinol) CoQ10 forms were measured simultaneously by high-performance liquid chromatography (HPLC) with electrochemical detection and a post-analytical reducing column using a Nanospace HPLC apparatus (Shiseido, Tokyo, Japan) at baseline and on weeks 4 and 8. Values were normalized for plasma total cholesterol and reported as CoQ10/total cholesterol (μmol/mol). Oxidized CoQ10 was expressed as percent of ubiquinone on total CoQ10 concentrations.

### 2.7. Determination of Serum Nitric Oxide Metabolites

Serum NO was indirectly determined in terms of its products, nitrite, and nitrate (NOx), by the Griess reaction as modified by Miranda et al. [34]. The method is based on a two-step process. The first step is the conversion of nitrate to nitrite using nitrate reductase; the second step is the addition of the Griess reagent (1% sulphanilamide and 0.1% N (-naphthyl) ethylenediamine), which converts nitrite into a deep-purple azo compound. This chromophore was measured colorimetrically at 540 nm. NOx were expressed as μm.

### 2.8. LDL Oxidation Assay

The peroxidation of serum lipid components was monitored by following the kinetic of conjugated dienes formation at 234 nm after exposure to copper sulfate. The method was optimized in a microplate reader (Synergy HT, Biotek Instruments, Winooski, VT, USA). Briefly, serum was centrifuged at 10,000 *g* for 2 min and 1:100 in PBS 5 mM sodium citrate 18 mM. Reaction was carried out at 37 °C and was started by the injection of CuSO4 freshly prepared at a final concentration of 25 μm in distilled water. Sigmoidal kinetic was recorded for 6 h every 5 min. The most representative indexes of sample resistance to peroxidative insult were calculated using GEN5 software version 2.0 (Biotek Instruments, Winooski, VT, USA): namely, the length of initiation phase (lag time), rate of dienes production (Vmax), and delta OD (optical density) corresponding to the blank subtracted plateau.

### 2.9. Sample Size and Statistical Analysis

Sample size determination was based on a previous meta-analysis of 5 randomized controlled trials (RCTs) evaluating the effects of CoQ10 on vascular endothelial dysfunction in humans [35]. In particular, the mean and standard deviation (SD) from one of the RCTs were used as reference values [24]. Considering a mean FMD change of +1.60 ± 1.16 in the treated groups and of −0.4 ± 2.24 in the placebo group, 14 subjects per group would be required to detect a difference with 80% power and a 5% two-sided type I error rate. Assuming a 20% dropout rate, a total number of 52.5 subjects were needed and a sample size of 51 subjects was deemed appropriate.

Descriptive statistics were used to summarize the study population. Data from continuous variables were expressed as mean (SD) and tested for normality using the Shapiro–Wilk test. Data on the primary endpoint followed a normal distribution, therefore parametric tests were used. Repeated measures ANOVA with Greenhouse–Geisser correction was used to assess differences between groups in continuous variables. Effect sizes for the primary endpoint were calculated using Cohen’s d statistic [36]. One-way ANCOVA was performed on the primary endpoint to correct for possible confounding factors. Pearson’s correlation was used to evaluate the association between endpoints. Statistical significance was defined as a two-sided *p* value < 0.05.

All statistical analyses were performed with SPSS version 25.0 (SPSS Inc., Chicago, IL, USA). Mediation analysis was performed using model 4 of the PROCESS Macro for SPSS with a bootstrapping procedure involving 10,000 re-samples to generate model estimates and confidence intervals. 

## 3. Results

### 3.1. Study Population

FMD was measured in 78 subjects who met the inclusion criteria; those with an FMD value of 2.5%–6% were enrolled in the study (*n* = 51). Three participants dropped out. The recruitment process and the reasons for withdrawal are summarized in the CONSORT flow chart in Figure 1.

In line with the protocol, dropouts were not removed from the ITT analysis and were not replaced. The PP therefore consisted of 48 subjects, 17 receiving ubiquinol 200 mg/day, 15 receiving ubiquinol 100 mg/day group, and 16 receiving a placebo. Their clinical and demographic characteristics are reported in Table 1. The three groups did not differ in age, gender, baseline BMI, biochemical variables, or lipid profile. No adverse events or major protocol deviations or violations were reported.

At the end of the study, no significant changes were detected among the three experimental groups in the plasma lipid profile (total cholesterol, LDL-C, HDL-C, and triglycerides) or in the other biochemical variables, i.e., hemoglobin, hematocrit, leukocyte count, ESR, ALT, creatinine, creatine kinase, glucose, HOMA index, and hs-CRP.

### 3.2. Flow-Mediated Dilation

The subjects randomized to the three treatment groups showed comparable baseline FMD (Table 2). According to repeated measures one-way ANOVA, at the end of the study (week 8), subjects in both treatment groups showed a significant FMD increase compared with those receiving the placebo (200 mg/day, +1.28% ± 0.90%; 100 mg/day, +1.34% ± 1.44%; placebo −0.41% ± 1.51%; F = 9.145; *p* < 0.001; Table 2).

Moreover, a statistically significant interaction was found between dosage and time on FMD (F(4, 90) = 4.663, *p* = 0.002, partial η_2_ = 0.172). According to the post-hoc comparisons, the differences in FMD increase between the groups receiving ubiquinol were not significant (*p* = 0.88), suggesting that its effect on FMD may be dose-independent. The effect remained significant after adjustment for age, gender, BMI, blood glucose, and LDL-C (one-way ANCOVA, r^2^ = 0.44, F = 8.502, *p* = 0.001). None of the clinical or demographic variables included in the multiple regression model displayed a significant interaction with the group variable.

As regarding the secondary endpoint, change in FMD on week 4, analysis of the effects of treatment did not highlight significant differences between the groups (F = 0.438; *p* = 0.648), whereas the increment from week 4 to week 8 was significant in both treated groups (F = 5.043; *p* = 0.011).

### 3.3. Plasma Total and Oxidized Coenzyme Q10

Total CoQ10 levels at baseline were comparable in the three groups (Table 3). Repeated measures one-way ANOVA, performed to compare plasma CoQ10 concentrations in the three groups and to evaluate participant compliance, demonstrated that ubiquinol supplementation induced a significant increase in plasma CoQ10 both at 4 (F = 36.551, *p* < 0.001) and 8 (F = 46.516, *p* < 0.001) weeks. The increments were dose-dependent (r = 0.80, F = 171.0, *p* < 0.0001), although the difference between 4 and 8 weeks was not significant in either group (paired t-test; 100 mg/day, *p* = 0.849; 200 mg/day, *p* = 0.879).

A highly significant and dose-dependent decrease in the percentage of oxidized CoQ10 was observed both at 4 weeks (F = 11.023, *p* < 0.001) and at 8 weeks (F = 10.087; *p* < 0.001), although the difference between 4 and 8 weeks was not significant (paired t-test; placebo, *p* = 0.183; 100 mg/day, *p* = 0.523; 200 mg/day, *p* = 0.356).

Data analysis highlighted a significant linear correlation between FMD values and plasma CoQ10 levels (r = 0.30, F = 13.698, *p* < 0.001; Figure 2A), but no correlation between FMD values and oxidized/total plasma CoQ10 (r = −0.14; F = 2.677, *p* = 0.104). Moreover, FMD changes showed a linear correlation with plasma CoQ10 (r = 0.33, F = 5.668, *p* = 0.022; Figure 2B) and an even stronger correlation with improvements in CoQ10 oxidative status (r = −0.35, F = 6.565; *p* = 0.014; Figure 2C).

### 3.4. Evaluation of Serum Nitric Oxide Metabolites

Serum NOx showed comparable levels among the groups at baseline (*p* = 0.969), whereas at the end of the study, it showed a significant increase (*p* = 0.016) in the treated groups compared with the placebo group (repeated measures ANOVA; 200 mg/die +9.3 ± 16.1 μm; 100 mg/die +5.9 ± 11.9 μm; placebo −4.9 ± 13.4 μm; Table 3). The dose-dependent increase of serum NO metabolites seen in the two treated groups was not significant (*p* = 0.496).

Analysis of baseline and T2 data demonstrated a significant positive correlation between serum NOx and FMD (r = 0.20, F = 3.931, *p* = 0.049), the FMD changes reflecting the NOx concentration (r = 0.36, F = 6.802, *p* = 0.012; Figure 3A). Moreover, the increase in total CoQ10 correlated with the increase in NOx bioavailability (r = 0.28, F = 4.127, *p* = 0.048), whereas an inverse correlation, found between the percentage of oxidized CoQ10 and NOx, was not significant (*p* = 0.09). A further exploratory analysis was performed without the data from the 20 subjects whose total CoQ10 levels at the end of the study exceeded 500 μmol/mol cholesterol, corresponding to a concentration of 2.5 μg/mL in subjects with a total cholesterol level of 220 mg/dL. The value of 500 μmol/mol was taken from previous studies, including one from our group, where it was identified as the dosage threshold conferring cardiovascular benefit on CVD patients [25]. It was thus reasonable to hypothesize that a saturating effect above this level should weaken the correlation. In fact, subgroup analysis showed that plasma CoQ10 correlated positively with circulating NOx concentrations (r = 0.30, F = 5.244, *p* = 0.026) and consequently with FMD values (r = 0.22, F = 4.308, *p* = 0.041).

### 3.5. Evaluation of Plasma Oxidized LDL and LDL Susceptibility to Oxidation In Vitro

Plasma oxLDL, measured at baseline and at the end of the study, showed no significant differences between the treatment groups (one-way ANOVA; T0, F = 1.845, *p* = 0.170; T2, F = 0.101, *p* = 0.905). The mean baseline concentration was 78.7 ± 17.3 U/L. A significant positive correlation was found between baseline oxLDL and LDL-C levels (r = 0.36, F = 6.650, *p* = 0.013). Ubiquinol supplementation did not exert significant effects on plasma oxLDL, either in terms of concentration (repeated measures ANOVA, F = 1.106, *p* = 0.340) or of the oxLDL/LDL-C ratio (F = 0.190, *p* = 0.827) (Table 3).

An in vitro assay of LDL susceptibility to oxidation measured as lag time (minutes) found no significant differences between the treated groups at baseline (one-way ANOVA, F = 1.076, *p* = 0.350). The mean baseline lag time was 108.3 ± 33.4 min. Supplementation with ubiquinol induced a significant increase in the 200 mg/day group (paired t-test; placebo, *p* = 0.156; 100 mg/day, *p* = 0.573; 200 mg/day, *p* = 0.017). The mean lag time increment in the 200 mg/day group was 16.0 ± 24.8 min; in these subjects, lag time positively correlated with the FMD values, with regard to both the single measurements (r = 0.53, F = 12.350, *p* = 0.001; Figure 3B) and the changes over the study course (r = 0.51, F = 5.665, *p* = 0.031). Moreover, a significant negative correlation was found between baseline lag time and percentage of oxidized CoQ10 (r = −0.38, F = 15.907, *p* < 0.0001). Finally, according to mediation analysis—performed to examine whether the beneficial effects of ubiquinol on endothelial dysfunction are mediated by its protective effects against LDL oxidation— a significant indirect effect of plasma CoQ10 on FMD through an enhanced antioxidant protection quantified as an increase in the oxidation lag time, ab (indirect effect) = 0.073, 95% BCa CI (0.007–0.141). The mediator lag time accounted for only 33.8% of the total effect (Figure 3C).

## 4. Discussion

The primary endpoint of the study was the effect of 8-week ubiquinol supplementation on EF in subjects with mild-to-moderate dyslipidemia, defined as an LDL-C concentration of 130–200 mg/dL, and moderate ED, defined as an FMD value of 2.5%–6%. None of the participants were on lipid-lowering drugs or supplements or showed clinical or biochemical signs of organ failure or CVD.

EF was assessed by measuring the FMD of the brachial artery, a reliable and accurate non-invasive ultrasound technique [37]. Baseline FMD was below 6% in approximately 75% of the 78 subjects screened for the study, reflecting a condition of moderate ED in our population. Since all efforts to standardize the technique have not yet produced a universally accepted FMD threshold for ED, a thorough analysis of the literature on the conditions strongly associated with ED, e.g., type 2 diabetes [38,39,40,41], and preliminary tests of healthy subjects aged 20–30 years by the experimental approach used in the present study allowed us to conclude that an FMD value exceeding 6% was a reasonable indicator of a healthy endothelium.

Ubiquinol treatment for 8 weeks induced a mean 1.3% FMD increase in both treated groups that was significant and did not seem to be dose-dependent. According to the latest consensus report, each 1% point increase in FMD involves a significant 8%–13% reduction in the risk of cardiovascular events [33].

In contrast, the plasma CoQ10 increase was significant and dose-dependent. Besides confirming compliance, these data showed that ubiquinol significantly improved oxidized CoQ10. Plasma CoQ10 reached a plateau already at 4 weeks and remained elevated until the end of the study. Interestingly, the EF improvement showed a different trend from plasma CoQ10, becoming detectable only at 8 weeks. The magnitude of the FMD improvement was comparable to that observed in healthy individuals or CVD patients undergoing exercise training [42,43], suggesting that ubiquinol supplementation could prove effective in primary and secondary prevention of CV diseases.

Ubiquinol did not ameliorate the lipid profile. This finding suggests that it plays a direct action by improving endothelium-dependent vasodilation independently of plasma cholesterol levels, which are known to be involved in the onset and progression of atherosclerosis. Our data partly contrast with those of a previous report describing a slight reduction in the levels of specific LDL subparticles after 2-week ubiquinol supplementation in healthy subjects [20]. However, a meta-analysis of randomized controlled trials assessing the effects of CoQ10 on the lipid profile concluded that of all lipid parameters, only plasma lipoprotein(a) is affected by CoQ10 administration [44].

The strong influence of ubiquinol on endothelium-dependent vasodilation is supported by the correlation found between its plasma levels and FMD. The positive correlation measured at baseline was maintained throughout the study. Remarkably, the increase in CoQ10 was proportional to the FMD increase. A strong correlation between improvement in CoQ10 oxidative status and FMD increments emphasizes the role of the reduced form of CoQ10 and the additional benefit of ubiquinol over the other ubiquinone [27].

To gain insight into the mechanisms involved in the ubiquinol-mediated amelioration of ED, FMD values were also tested against serum NOx concentrations. Their strong positive correlation supports the view that ubiquinol supplementation enhanced NO bioavailability, which is essential for EF and prevents the remodeling of small resistance arteries [45]. Moreover, further close correlations were found among the three main study endpoints, i.e. CoQ10, FMD, and NO.

Although the role of CoQ10 has extensively been explored in healthy subjects and in different pathological conditions, its effects on EF have mainly been described in patients with CVD and related chronic states such as type 2 diabetes and CAD [46,47]. In a study involving CoQ10 supplementation in healthy non-smokers with hypercholesterolemia, Raitakari et al. found baseline FMD values that were comparable to those observed in our population, lending support to the notion that hypercholesterolemia alone is sufficient to induce ED in the absence of other risk factors. In line with their results, we found that treatment for 4 weeks was not sufficient to confer benefits on endothelium-dependent vasodilation [26]. Two studies conducted by our group in patients with chronic heart failure and CAD showed that 300 mg ubiquinone a day for 8 weeks improved cardiac performance and FMD of the brachial artery, highlighting its role on endothelial homeostasis [25,48]. In addition, we showed that the effect of CoQ10 was also mediated by an increase in endothelium-bound extracellular superoxide dismutase (ecSOD), a key enzyme that protects against the detrimental action of superoxide anion on the arterial wall [25]; in this study, we were also able to induce in vivo a beneficial effect on endothelial cells that mirrored in vitro evidence supporting an anti-inflammatory and antiaging effect of ubiquinol [49].

Surprisingly, ubiquinol supplementation induced no changes in circulating oxLDL, probably as a result of the relatively low baseline levels. However, a significant reduction in LDL susceptibility to oxidation in vitro was achieved in subjects treated with the higher ubiquinol dosage, in agreement with Raitakari et al. Mediation analysis indicated that this effect accounts for a third of the total effect of CoQ10 on FMD; the remaining quota is probably to be ascribed to mechanisms not addressed in this study, like improved ecSOD activity [25].

Our study has several strengths, including the evaluation of plasma CoQ10 at both time points, which allowed demonstrating participant compliance. Moreover, we closely followed the latest FMD assessment guidelines [33], including the requirement that FMD be measured by the same trained sonographer, the use of a stereotactic probe-holder to maintain the same scan throughout the examination, and the continuous measurement of post-deflation diameter, to capture the actual brachial artery peak diameter. Some limitations also need to be discussed. First, the limited population studied could not allow to draw firm conclusions on the beneficial effect of ubiquinol. However, we reached the sample size of 51 subjects required by the power analysis performed a priori on the primary endpoint, with a final drop-out rate even below the initially expected 20%. Moreover, a large Cohen’s effect size was achieved for both 200 mg/day (0.89) and 100 mg/day (1.30) groups. Second, we did not evaluate endothelial-independent vasodilation, but only FMD. This prevents establishing whether the ED of our population was related to reduced NO bioavailability or to a functional defect in vascular smooth muscle cells. This choice was based on the relative lack of reproducibility of GTN-mediated vasodilation [50], and to our previous report on CVD patients showing no significant effect of CoQ10 on endothelium-independent vasodilation [25]. To address this problem and confirm the endothelium-dependent nature of FMD, NO was assessed at baseline and at the end of the study.

This is the first evidence of the protective effect of ubiquinol towards endothelial function in adult subjects with mild-to-moderate dyslipidemia and no other evidence of disease. Our results extend the previous evidence on the beneficial role of CoQ10 in CVD to a cohort of healthy subjects with risk factors and support the role of non-invasive imaging methods in the assessment of atherosclerosis at the subclinical stage. Given the strong relationship between ED and total CV risk, supplementation with ubiquinol could represent a feasible non pharmacologic approach to reduce the risk of cardiovascular events in a great number of subjects with subclinical atherosclerosis.

## 5. Conclusions

In conclusion, in healthy individuals with mild-to-moderate dyslipidemia, ubiquinol supplementation enhances endothelium-dependent vasodilation as assessed by FMD. This effect is related to improvements in oxidized CoQ10, to increased NO bioavailability, and partly to enhanced LDL antioxidant defences.

## Figures and Tables

**Figure 1 nutrients-12-01098-f001:**
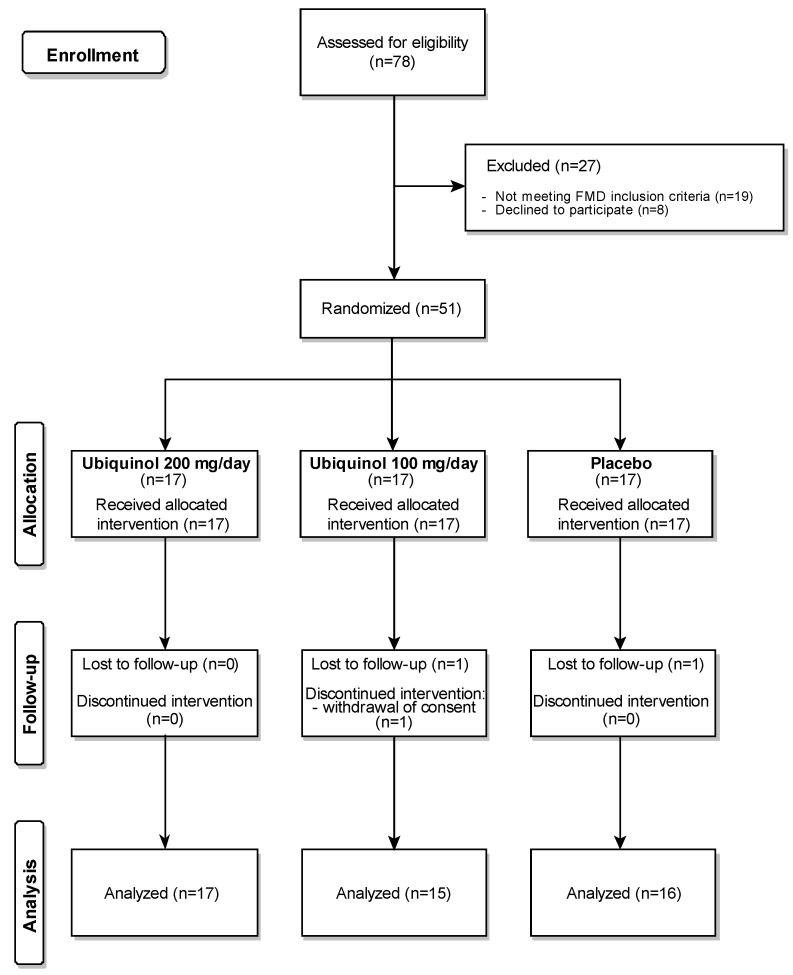
Consolidated Standards of Reporting Trials (CONSORT) flowchart. A total number of 78 patients were screened. Of these, 51 were randomized to the three groups and 48 completed the study. FMD, flow-mediated dilation.

**Figure 2 nutrients-12-01098-f002:**
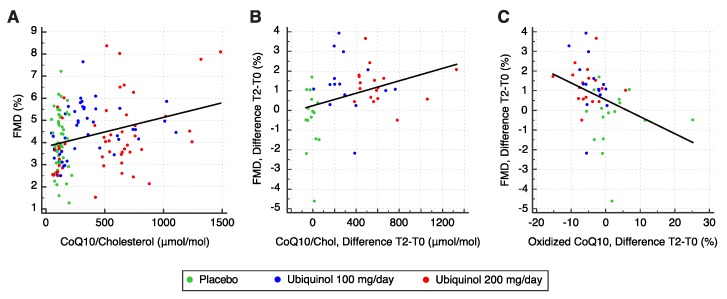
(**A**) Scatterplot showing the correlation between plasma coenzime Q10 (CoQ10) and flow-mediated dilation (FMD) for the pooled samples (*n* = 48 at each timepoint). (**B**,**C**) Scatterplots showing the correlations between the changes in FMD and (**B**) plasma CoQ10 or (**C**) oxidized CoQ10 at the three timepoints (*n* = 48).

**Figure 3 nutrients-12-01098-f003:**
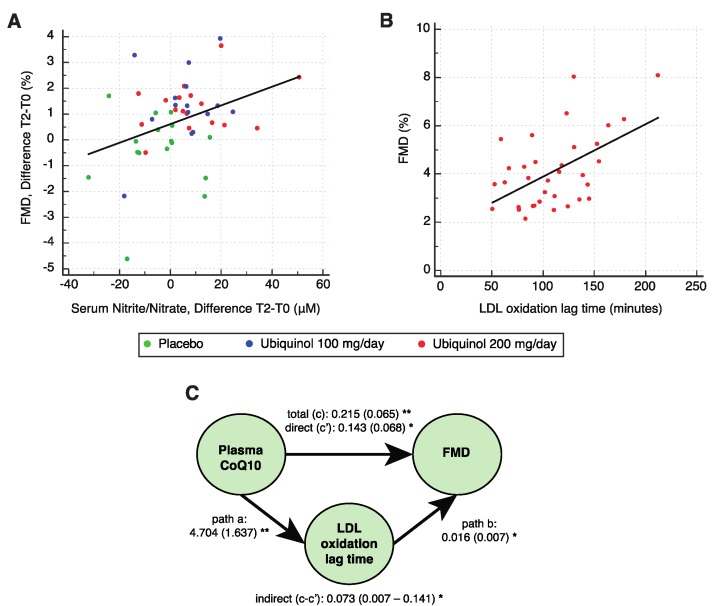
(**A**) Scatterplot showing the correlation between serum nitric oxide (NO) and FMD variations at the three timepoints (*n* = 48). (**B**) Scatterplot showing the correlations between low-density lipoprotein (LDL) oxidation lag time and FMD in the ubiquinol 200 mg/day group. (**C**) Results of mediation analysis of the effects of plasma CoQ10 on FMD mediated by LDL susceptibility to oxidation. *, *p* < 0.05; **, *p* < 0.01 for standardized bootstrapped (10,000 samples) total, direct, and indirect effect size. In brackets, are reported the standard errors for total and direct effects, and the 95% CI for indirect effect. The mediation procedure is described in Methods, paragraph 2.9.

**Table 1 nutrients-12-01098-t001:** Baseline clinical and demographic characteristics of the 48 subjects in the per-protocol population ^1^.

Variable	Ubiquinol, 200 mg/day (*n* = 17)	Ubiquinol, 100 mg/day (*n* = 15)	Placebo (*n* = 16)
Age (years)	58.2 (5.1)	59.3 (6.4)	59.6 (3.8)
Gender (males)	7	5	9
BMI (Kg/m^2^)	23.8 (3.2)	24.6 (3.6)	24.1 (3.4)
Heart rate (bpm)	64 (8)	69 (7)	67 (10)
Systolic blood pressure (mmHg)	128 (15)	129 (13)	125 (12)
Diastolic blood pressure (mmHg)	79 (7)	82 (9)	83 (10)
Hemoglobin (mg/dL)	14.0 (1.0)	14.6 (1.1)	14.6 (1.0)
WBC (cells/mm^3^)	5.97 (0.98)	7.08 (0.98)	6.67 (1.55)
Total cholesterol (mg/dL)	214.5 (33.3)	218.1 (29.4)	229.8 (31.9)
LDL cholesterol (mg/dL)	145.3 (25.7)	147.0 (26.3)	157.7 (29.4)
HDL cholesterol (mg/dL)	59.3 (15.9)	61.1 (14.8)	62.6 (17.3)
Triglycerides (mg/dL)	105.2 (59.1)	109.7 (50.0)	99.4 (70.4)
Glucose (mg/dL)	95.2 (9.2)	98.3 (9.3)	96.7 (11.0)
HOMA index	1.36 (0.65)	1.12 (0.49)	1.33 (0.57)
Creatinine (mg/dL)	0.84 (0.15)	0.89 (0.23)	0.86 (0.16)
Alanine aminotransferase (U/L)	21.0 (9.9)	18.3 (9.9)	18.6 (6.4)
Creatine kinase (U/L)	108.3 (32.3)	112.0 (45.1)	365.5 (1009.5)
hs-CRP (mg/L)	0.26 (0.27)	0.11 (0.09)	0.22 (0.27)
ESR (mm/h)	15.1 (10.2)	16.5 (10.1)	10.8 (6.5)

^1^ Data are mean (standard deviation) for continuous variables. BMI, body mass index; WBC, white blood cells; LDL, low-density lipoprotein; HDL, high-density lipoprotein; HOMA, homeostatic model assessment; hs-CRP, high sensitivity C-reactive protein; ESR, erythrocyte sedimentation rate.

**Table 2 nutrients-12-01098-t002:** Summary of the primary endpoint: Change in flow-mediated dilation (FMD) on week 8, and of the secondary endpoint; Change in FMD on week 4, assessed in the per-protocol population (*n* = 48) ^1^.

Variable	Ubiquinol, 200 mg/day (*n* = 17)	Ubiquinol, 100 mg/day (*n* = 15)	Placebo (*n* = 16)
Baseline brachial artery diameter (mm)	4.02 (1.0)	3.97 (0.60)	4.19 (1.02)
**Flow-mediated dilation (%)**			
T0 (recruitment)	3.48 (1.12)	3.80 (0.95)	4.06 (1.13)
T1 (week 4)	4.34 (1.81)	4.63 (0.90)	4.51 (1.54)
T1-T0 difference	0.86 (1.62)	0.84 (1.16)	0.45 (1.37)
T2 (week 8)	4.75 (1.68)	5.14 (1.12)	3.65 (1.06)
T2-T0 difference **	1.28 (0.94)	1.34 (1.44)	−0.41 (1.51)
T2-T1 difference *	0.41 (1.48)	0.51 (0.96)	−0,86 (1.52)
Effect size (Cohen’s d, 95% CI)	0.89 (0.56–1.40)	1.30 (0.38–2.09)	−0.37 (−1.10–0.20)

^1^ Data are mean (standard deviation). T, time; CI, confidence interval. *, *p* < 0.05; **, *p* < 0.01 for two-way repeated measures analysis of variance (ANOVA).

**Table 3 nutrients-12-01098-t003:** Summary of the secondary endpoints (in bold) assessed in the per-protocol population (*n* = 48) ^1^.

Variable	Ubiquinol, 200 mg/day (*n* = 17)	Ubiquinol, 100 mg/day (*n* = 15)	Placebo (*n* = 16)
**Plasma CoQ10 (μmol/mol)**			
T0 (recruitment)	108.7 (28.9)	145.5 (80.3)	131.1 (28.9)
T1 (week 4)	712.0 (262.6)	449.7 (310.1)	120.7 (32.4)
T1-T0 difference ***	603.3 (247.1)	304.2 (256.6)	−10.4 (28.3)
T2 (week 8)	723.0 (262.8)	461.9 (245.9)	121.2 (50.1)
T2-T0 difference ***	614.4 (244.8)	316.5 (202.5)	−9.9 (37.2)
T2-T1 difference	11.1 (150.5)	12.3 (180.6)	0.5 (35.6)
**Oxidized CoQ10 (%)**			
T0 (recruitment)	12.6 (5.9)	11.7 (4.7)	11.6 (4.8)
T1 (week 4)	6.0 (1.9)	8.4 (4.2)	11.1 (5.1)
T1-T0 difference ***	−6.6 (4.6)	−3.3 (3.8)	−0.5 (2.5)
T2 (week 8)	7.2 (6.8)	7.6 (2.4)	14.1 (8.2)
T2-T0 difference ***	−5.4 (4.4)	−4.0 (3.1)	2.5 (7.4)
T2-T1 difference	1.2 (6.1)	−0.7 (2.8)	3.0 (7.3)
**Serum nitric oxide metabolites (μm)**			
T0 (recruitment)	63.4 (16.6)	63.7 (16.0)	64.8 (15.3)
T2 (week 8)	72.8 (20.0)	69.5 (14.8)	59.9 (15.6)
T2-T0 difference *	9.3 (16.1)	5.9 (11.9)	−4.9 (13.4)
**Plasma oxidized LDL (U/L)**			
T0 (recruitment)	74.0 (21.1)	77.2 (16.4)	85.2 (12.1)
T2 (week 8)	82.6 (17.2)	79.7 (17.2)	80.2 (25.1)
T2-T0 difference	8.6 (26.3)	2.5 (23.5)	−5.0 (28.4)
**LDL oxidation lag time (min)**			
T0 (recruitment)	102.4 (30.2)	104.3 (37.9)	118.2 (31.9)
T2 (week 8)	118.4 (43.2)	100.7 (37.4)	129.1 (40.1)
T2-T0 difference *	16.0 (24.8)	−3.6 (24.3)	11.0 (29.4)

^1^ Data are mean (standard deviation). *, *p* < 0.05; ***, *p* < 0.001 for two-way repeated measures ANOVA. CoQ10, coenzyme Q10. The bold indicate each endpoint.

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
