# Peer review of "Ubiquinol Ameliorates Endothelial Dysfunction in Subjects with Mild-to-Moderate Dyslipidemia: A Randomized Clinical Trial"

_nutrients, 2020, doi:10.3390/nu12041098_

Round 1

Reviewer 1 Report

Comments to Authors:

Jacopo Sabbatinelli et al. showed ubiquinol reduced the endothelial dysfunction in dyslipidemia patients.
The manuscript has some issues in preparation and study design.

Major Comments:

1. Introduction: The introduction section needs to thoroughly rewrite to modify the structure.

i) First paragraph needs to start with CVDs demographical statistical information following describes what is dyslipidemia? other causes? readers need to understand very well1
ii) There is no description of Ubiquinol? Are there any in vivo in vitro studies on ubiquinol in works of literature?
iii) what rationale to choose this study?

2.Methods:

i) what rationale to selected the ubiquinol dose 100,200 mg/day?

ii) The study design has some limitations: You should need to redesign include more clinical parameters such as insulin resistance, HOMA-IR, ACE,iNOS,eNOS, glyoxalase, etc.?

iii) Does the study need to include in slico tools to analyze prediction in patients' database?

3. Do you check any nitrosative markers? reactive nitrogen species(RNS) levels?

Minor Comments:

Please revise the manuscript carefully free from any grammatical, spelling mistake?

Author Response

We are grateful to the Reviewer for the accurate analysis of our work and for the constructive comments and suggestions. All changes are highlighted in red in the revised text.

Major Comments:

1. Introduction: The introduction section needs to thoroughly rewrite to modify the structure.

i) First paragraph needs to start with CVDs demographical statistical information following describes what is dyslipidemia? other causes? readers need to understand very well

We added epidemiological information on dyslipidemia to first paragraph of the introduction.

ii) There is no description of Ubiquinol? Are there any in vivo in vitro studies on ubiquinol in works of literature?

We added further information on ubiquinol in the introduction, also to clarify the rationale of our study.

iii) what rationale to choose this study?

Following your suggestion, the rationale of the study has been better outlined in the revised introduction.

2.Methods:

i) what rationale to selected the ubiquinol dose 100,200 mg/day?

A dosage of 200 mg per day is the highest amount allowed in Italy and in several EU countries as a nutritional supplement. Given that ubiquinol target population is composed mainly by healthy subjects, we considered the maximum dosage, and half of the maximum dosage, as a good reference. Moreover, 100 mg is the ubiquinol dosage contained in many commercial formulations. Finally, considering the higher bioavailability compared to the oxidized form ubiquinone, these dosages were able to produce similar plasma levels of the ones measured in our previous studies using 300 mg ubiquinone in CVD patients (Tiano et al. 2007, 10.1093/eurheartj/ehm267).

ii) The study design has some limitations: You should need to redesign include more clinical parameters such as insulin resistance, HOMA-IR, ACE,iNOS,eNOS, glyoxalase, etc.?

With respect to the limited time provided for the present revision, we added the HOMA index value, which was calculated based on the acquired data. Unfortunately, we are unable to run further experiments on the plasma samples, since this would have required to re-conduct the study. However, we will take into account your valuable suggestions for developments of this study.

iii) Does the study need to include in slico tools to analyze prediction in patients' database?

We do not think that in silico studies are required. The effect of ubiquinone was already tested in vivo in respect to CVD and endothelial function. The aim of the study was to evaluate the efficacy in the general population with mild risk factors such as hypercholesterolemia.

3. Do you check any nitrosative markers? reactive nitrogen species(RNS) levels?

We did measure NO levels as nitrites and nitrates. It could have been useful to study protein nitrosylation, but we had to make a selection of the most relevant markers according to the budget of the study. We will consider it for further developments.

Minor Comments:
Please revise the manuscript carefully free from any grammatical, spelling mistake?

The manuscript was revised by the authors and by a professional editing service to correct any mistake.

Reviewer 2 Report

In this study, Sabbatinelli et al, are studying the effect of Ubiquinol on endothelial function using a randomized double-blind single center trial in patients with moderate LDL cholesterol level and with no clinical sign of cardiovascular diseases. They showed that ubiquinol significantly improved endothelial function by improving nitric oxide bioavailability and partly by enhancing LDL antioxidant protection. This study is very exciting and very well designed; however, there is some minor concern that are listed below.

Minor concern

  1. What is the rationale behind using only male and female with post-menopause? This point is not addressed anywhere in the paper.
  2. How the dose of 200 mg/day or 100 mg/day was determined? And why authors used 2 different doses?
  3. Authors should clarify why they used 8 weeks as treatment period?
  4. The authors need to discuss further why they did not see reduction in oxLDL.
  5. The authors stated that FMD improvement is partly mediated by enhanced LDL antioxidant protection; however, treatment did change oxLDL. Please clarify

Author Response

We are grateful to the Reviewer for the accurate analysis of our work, for the appreciation, and for the constructive comments and suggestions. All changes are highlighted in red in the revised text.

Minor concern

1. What is the rationale behind using only male and female with post-menopause? This point is not addressed anywhere in the paper.

Premenopausal women were excluded given the high variability of FMD during the different phases of the menstrual cycle. We clarified this point in the methods section. Moreover, in designing the inclusion criteria, we accounted for the different level of CV risk between same-aged males and females.

2. How the dose of 200 mg/day or 100 mg/day was determined? And why authors used 2 different doses?

A dosage of 200 mg per day is the highest amount allowed in Italy and in several EU countries as a nutritional supplement. Given that the target population is composed mainly by healthy subjects, we considered the maximum dosage, and half of the maximum dosage, as a good reference. Moreover, 100 mg is the ubiquinol dosage contained in many commercial formulations. Finally, considering the higher bioavailability compared to the oxidized form ubiquinone, these dosages were able to produce similar plasma levels of the ones measured in our previous studies using 300 mg ubiquinone in CVD patients (Tiano et al. 2007, 10.1093/eurheartj/ehm267)

3. Authors should clarify why they used 8 weeks as treatment period?

Two weeks is the time needed to reach the plateau of plasma CoQ levels. The analysis has been conducted for a longer period in order to produce a tangible biological effect. In particular, we used previous studies assessing FMD modulation by Q10 as a reference (Tiano et al. 2007, 10.1093/eurheartj/ehm267; Raitakari et al. 2000, 10.1016/s0891-5849(00)00201-x). In order to increase the observation time frame, we evaluated selected endpoint over both a 4- and 8-week time span.

4. The authors need to discuss further why they did not see reduction in oxLDL.

Indeed, it was quite surprising for us not to observe any significant decrease in the plasma LDL oxidative status. Notably, healthy volunteers, differently from disease patients, did not show a significant baseline level of LDL oxidation. Thus, we reasoned that no significant improvement could have been expected. On the other hand, this observation supports the conclusion that ubiquinol exerts is able to enhance endothelial function beyond its effect as antioxidant in plasma lipoproteins. In other words, the improvement in FMD is not simply related to an indirect effect triggered by the improvement of the oxidative status of lipoproteins.

5. The authors stated that FMD improvement is partly mediated by enhanced LDL antioxidant protection; however, treatment did change oxLDL. Please clarify.

The statement refers to the mediation analysis, reported in Figure 3C, showing that one third of the linear correlation between plasma CoQ10 and FMD is mediated by the variable ‘LDL oxidation lag time’. This variable refers to the results of the in vitro assessment of LDL susceptibility to oxidation by copper. Even if plasma levels of oxLDL were unchanged (see above), we still think that the in vitro evidence support the hypothesis that ubiquinol treatment confers a certain degree of protection to LDL against further oxidation.

Round 2

Reviewer 1 Report

The authors addressed all my comments.

Author Response

We are grateful to the Reviewer.